# Efficacy and Safety Analysis of Nab-Paclitaxel Treatment in Elderly Patients with HER-2 Negative Metastatic Breast Cancer: NEREIDE Study

**DOI:** 10.3390/cancers17132069

**Published:** 2025-06-20

**Authors:** Giuseppina Rosaria Rita Ricciardi, Alessandro Russo, Maria Vita Sanò, Angela Prestifilippo, Antonio Russo, Vittorio Gebbia, Livio Blasi, Dario Giuffrida, Giuseppa Scandurra, Antonio Savarino, Alfredo Butera, Nicolò Borsellino, Francesco Verderame, Michele Caruso, Vincenzo Adamo

**Affiliations:** 1Department of Onco-Hematology, Papardo Hospital, 98158 Messina, Italy; alessandro.russo@hunimed.eu (A.R.); vadamo@unime.it (V.A.); 2Department of Medical Oncology, Humanitas Istituto Clinico Catanese, 95045 Catania, Italy; mariavita.sano@humanitascatania.it (M.V.S.); michele.caruso@humanitascatania.it (M.C.); 3Department of Biomedical Sciences, Humanitas University, 20027 Milan, Italy; 4Medical Oncology Unit, Istituto Oncologico del Mediterraneo Hospital, 95029 Catania, Italy; angela.prestifilippo83@gmail.com (A.P.); dgiuff57@gmail.com (D.G.); 5Department of Surgical-Oncological- and Oral Sciences-Section of Medical Oncology, University of Palermo, 90128 Palermo, Italy; antonio.russo@usa.net; 6Medical Oncology Unit, CdC Torina, 90146 Palermo, Italy; vittorio.gebbia@unikore.it; 7Department of Medicine and Surgery, Kore University of Enna, 94100 Enna, Italy; giuseppa.scandurra@unikore.it; 8Medical Oncology Unit, ARNAS Civico Hospital, 90127 Palermo, Italy; livio.blasi61@gmail.com; 9Medical Oncology Unit, Cannizzaro Hospital, 95100 Catania, Italy; 10Medical Oncology Unit, Barone Lombardo Hospital, 92024 Canicattì, Italy; antonio.savarino@aspag.it; 11Medical Oncology Unit, San Giovanni di Dio Hospital, 50143 Agrigento, Italy; alfredo.butera@aspag.it; 12Medical Oncology Unit, Buccheri La Ferla Fatebenefratelli Hospital, 90123 Palermo, Italy; borsellino.nicolo1962@gmail.com; 13Medical Oncology Unit, Cervello Hospital, 90146 Palermo, Italy; francescoverderame1@gmail.com

**Keywords:** metastatic breast cancer, elderly, nab-paclitaxel

## Abstract

NEREIDE was an observational, retrospective, multicenter study carried out in 11 Sicilian oncology centers, evaluating the safety and activity of nab-paclitaxel (nab-P) in HER2-negative mBC patients who are ≥65 years old. The results of this study further confirm the efficacy and safety of single-agent nab-P in pretreated, elderly, HER2-negative mBC in a real-world setting.

## 1. Introduction

Age is the leading risk factor for breast cancer, as the risk of developing breast cancer increases with age, with a 10-year probability of 3% of developing breast cancer in women over 50 years of age and more than 4% after 70 years [1]. The incidence of breast cancer in elderly women is increasing, and patients aged 65 years or older account for ~45% of new cases [2].

Advanced age at diagnosis is associated with more favorable biological factors, such as high hormone receptor (HR) expression, decreased Human Epidermal Growth Factor Receptor 2 (HER2/c-erbB2) overexpression, low tumor grading, and low proliferative index [3,4,5].

Elderly patients are usually associated with worse outcomes for multiple reasons: under treatment (omission of treatment and/or inappropriate treatment), lower rates of treatment compliance, lower adherence rates, higher rates of lower intensity treatment, more toxicity with treatment, and more non-guideline-recommended treatments [6,7]. Elderly patients with breast cancer include a heterogeneous population with both fit, frail, and vulnerable subjects that require different management [8]. Screening tools for a complete geriatric assessment allow a correct evaluation of health status. They should be applied in clinical practice to select the best treatment strategy for these patients [9,10,11,12]. Indeed, a multi-dimensional geriatric assessment improves compliance and tolerability of treatments, as well as quality of life and survival [13]. However, these tools are often underused in clinical practice. International guidelines recommend that elderly women with metastatic breast cancer be treated as young patients, reserving a different approach only for non-fit older patients [14,15]. Taxanes are one of the most effective cytotoxic drugs for treating mBC, either as single agents or in combination. They are associated with a survival benefit greater than other chemotherapeutic agents [16,17]. Despite their clinical activity, taxanes may be limited by their safety profile and side effects, such as hypersensitivity reactions and peripheral neuropathy, that still represent major issues. Moreover, corticosteroid and antihistamine premedication before taxane administration is mandatory but causes additional side effects [18,19,20]. Pharmacological studies reported a decreased clearance of both paclitaxel and docetaxel in elderly patients compared to non-elderly patients, suggesting safer use of the lower range of proposed doses of the weekly regimens in elderly patients. However, despite a lower incidence of dose-limiting toxicity for 3-weekly taxanes, severe neutropaenia, the onset of neuropathy (paclitaxel), or fatigue and fluid retention (docetaxel), even with weekly regimens, can be troublesome and eventually require dose modifications [21].

Nanoparticle albumin-bound paclitaxel (nab-paclitaxel, nab-P) is a solvent-free colloidal suspension of paclitaxel and human serum albumin, developed to exploit the anti-tumor activity of conventional paclitaxel with a shorter infusion schedule (30 min vs. 3 h) and without the need for premedication. Nab-P demonstrated higher antitumor activity in a large, randomized phase III trial comparing 3-week nab-paclitaxel 260 mg/m^2^ and 3-week paclitaxel 175 mg/m^2^. This study reported a significant overall response rate (ORR) advantage with nab-paclitaxel compared to solvent-based paclitaxel (33% vs. 19%, *p* = 0.001). This benefit was observed in both treatment-naïve (ORR 42% with nab-paclitaxel vs. 27%, *p* = 0.029) and pretreated patients (ORR 27% vs. 13%, *p* = 0.006). Interestingly, the ORR benefit favoring nab-paclitaxel was also seen in a subgroup of patients with ≥65 years (ORR of 34% vs. 19%, *p* = 0.001). Moreover, PFS was significantly longer in the nab-paclitaxel arm (23 weeks vs. 16.9 weeks, *p* = 0.006), with a survival trend also seen in the nab-P arm (65.0 vs. 55.7 months, *p* = 0.046) [22]. Another phase II trial reported a statistical and clinically significant advantage in terms of PFS (exceeding 5 months) in first-line mBC patients receiving weekly nab-paclitaxel vs. the highest standard dose of docetaxel [23].

Data on the efficacy and safety of nab-paclitaxel in elderly patients come mostly from retrospective analyses. A post hoc analysis of two phase II and III studies [22,23] was conducted to evaluate the efficacy and safety of weekly or every 3-week nab-P in elderly patients and compare two different nab-paclitaxel schedules with paclitaxel or docetaxel in the subgroup of patients aged ≥65 years [24], demonstrating that the use of the weekly nab-paclitaxel is effective and safe in elderly and represent the best therapeutic schedule in this setting. Two prospective studies evaluated the role of nab-P in elderly mBC: The phase II EFFECT study and the real-world observational study ABREAST. In the EFFECT study, nab-P at 100 mg/m^2^ was significantly better tolerated than 125 mg/m^2^ weekly in older patients with mBC as a first-line regimen [25]. In the ABREAST study, the real-world PFS of nab-P was similar in mBC patients aged < and ≥70 years [26].

Nowadays, nab-paclitaxel is steadily part of the therapeutic armamentarium of mBC either as a single agent or in combinations, and, given the increased incidence of breast cancer in the elderly, further characterization of the safety and activity profile of this agent in this population is needed.

In this retrospective multicenter study, we further explored the safety and efficacy of nab-paclitaxel in HER-2-negative mBC patients aged ≥65 years.

## 2. Patients and Methods

NEREIDE (Nab-paclitaxel in mEtastatic bREast cancer In ElDerly patiEnts) was an observational retrospective multicenter study designed to evaluate, in a real-life setting, the efficacy and safety of nab-paclitaxel treatment in HER-2-negative mBC elderly patients. The study design and major patient characteristics are summarized in Figure 1.

A total of 70 patients with HER2-negative metastatic breast cancer who were ≥65 years old were evaluated. The primary endpoint was the evaluation of the safety of nab-paclitaxel. Secondary endpoints were the percentage of objective responses (ORRs), the progression-free survival (PFS), and the overall survival (OS). The inclusion criteria included age ≥65 years; diagnosis of metastatic disease confirmed histologically or cytologically; availability of the following biological parameters: tumor grading, estrogen receptor (ER), progesterone receptor (PgR), and HER-2 status; HER-2 negative disease, defined as an immunohistochemistry score of 0–1 or an immunohistochemistry (IHC) score of 2 without gene amplification by fluorescence in situ hybridization (FISH); patients with bilateral breast cancer, if both tumors were negatively affected by HER-2; availability of the paraffin block relative to the primary tumor and/or the one related to biopsies of metastatic disease; any chemotherapeutic treatment and/or hormonal treatment for advanced or locally advanced disease; patients who had received treatment with nab-paclitaxel in the metastatic setting both with the three-weekly schedule and with the weekly schedule; patients with at least one measurable target lesion documented by computed tomography (CT) scan or magnetic resonance imaging (MRI) according to the Response Evaluation Criteria in Solid Tumors (RECIST) v1.1; and a description of adverse events reported with nab-paclitaxel using National Cancer Institute (NCI) Common Terminology Criteria for Adverse Events CTCAE v4.0 criteria.

Patients with the following characteristics were excluded: (1) age less than 65 years; (2) HER-2 positive metastatic disease, defined as an IHC score of 3 or gene amplification by FISH; (3) had not received treatment with nab-paclitaxel in the metastatic setting; and (4) the presence of symptomatic brain metastases. The local Ethical Committee approved the study.

Statistical analysis. The correlation between the efficacy of nab-paclitaxel and the different clinical–pathological variables was performed using the chi-square test. Univariate and multivariate analyses estimated the association between clinical–pathological variables and clinical outcomes. The univariate and multivariate analysis was performed using the Cox regression model, and the results were considered statistically significant if a *p*-value < 0.05 was reached. Safety was evaluated using the National Cancer Institute (NCI) Common Terminology Criteria for Adverse Events (CTCAE) version 4.0. The tumor response was assessed according to the RECIST criteria version 1.1 [27]. The ORR was defined as the sum of complete and partial responses (CR + PR). Progression-free survival (PFS) was defined as the time between nab-P treatment initiation and the date of the first evidence of disease progression. Overall survival (OS) was defined as the time between nab-P treatment initiation and the date of patient death for any cause. The PFS and OS curves were estimated using the Kaplan–Meier method. Statistical analyses were performed using SPSS v17.0 for Windows.

## 3. Results

The present study included seventy HER-2 negative metastatic breast cancer patients aged ≥65 years treated with nab-paclitaxel. The study population had a median age of 67 (range 65–83) and a median Eastern Cooperative Oncology Group (ECOG) performance status (PS) of 1 (range 0–2). The major clinic–pathological characteristics of the patients are summarized in Table 1.

The different intrinsic molecular subtypes were represented as follows: Luminal A (18.8%), Luminal B HER-2 negative (62.5%), and triple negative (18.8%). The main sites of metastatic spread of the disease were 31% visceral and bone metastases, 16% bone metastases, 10% lung metastases, 10% visceral and lymph node metastases, and 6% liver metastases [Figure 2].

We included a heavily pretreated population, with 33% of the patients receiving nab-paclitaxel as a fourth or subsequent therapeutic line for metastatic disease. The distribution of patients based on previous line(s) of treatment received is summarized in Table 1.

Treatment schedule selection was made on treating physician preference. Most patients (87.1% of patients) received nab-paclitaxel at a dose of 260 mg/m^2^ every 3 weeks, whereas 12.9% were treated with weekly nab-paclitaxel at 125 mg/m^2^. Overall, 80.7% of patients had received previous taxane treatment in the neoadjuvant (28.6%), adjuvant (25.7%), or metastatic setting (26.2%). All 70 patients included in the study were evaluable for efficacy and safety.

The median number of nab-paclitaxel cycles received was six (range 1–21): 35.5% of patients received a dose reduction and 11.5% discontinued treatment because of adverse events. Grade 2/3 adverse events were observed in 47% of patients. The major toxicities reported were fatigue (61.5%), peripheral neuropathy (53.8%), and neutropenia (39.1%) [Table 2]. A higher incidence of AEs was reported with the q3w schedule (AEs observed in 85.7% of patients). G ≥ 3 AEs were reported in 17% of the patients and included leukopenia G3 (7%), asthenia G3 (4%), alopecia G3 (3%), nausea (1.5%), and peripheral neuropathy (1.5%). No grade 4 or 5 events were reported. All the grade 3 AEs were collected in patients treated with the three-weekly schedule. No treatment-related deaths were reported.

In terms of efficacy, an ORR of 31.3% was reported, including complete response (CR) in 6.3% and partial response (PR) in 25%. Stable disease (SD) was observed in 39.1% of patients treated with nab-paclitaxel, with an overall disease control rate (DCR) of 70.4%.

A median PFS of 6 months (95% CI, 2–38) was observed in the study population (Figure 3). Finally, a median OS of 40.5 months (95% CI 7–255) was observed (Figure 4).

## 4. Discussion

Elderly women represent a substantial and increasing population of patients with breast cancer [28]. Over 40% of new cases of breast cancer are diagnosed in women aged 65 and over [29]. However, this growing population of breast cancer patients is still underrepresented in clinical trials, causing a lack of data on the best therapeutic strategy in this subset of patients. Indeed, “strong” levels of evidence (level of evidence 1) on the treatment of such patients are lacking, due to the absence of dedicated clinical trials. To date, treatment of elderly patients is primarily based on data from retrospective analyses on a small percentage of elderly patients or from extrapolation of results from younger patient populations. As a result, elderly patients are often under-treated or treated inappropriately. The International Society of Geriatrics Oncology (SIOG) and European Society of Breast Cancer Specialists (EUSOMA) task force for the management of elderly breast cancer patients and the fifth ESO-ESMO international consensus guidelines for advanced breast cancer (ABC 5) clearly stated that age alone should not be considered the sole determinant in the therapeutic choice of such patients [15,16,30]. Furthermore, there is a limited representation of geriatric oncology across cancer guidelines, prompting greater consideration of older adults’ unique needs [31]. We know that most of the breast cancers that develop in elderly patients are luminal-type or with no expression/amplification of the oncogene c-erbB2. Such patients may be candidates for receiving chemotherapy, including taxanes, if they have a hormone receptor-negative (HR-) disease, a HR + disease that is refractory to endocrine therapy, or a HR + disease that progresses rapidly; in such cases, they may require chemotherapy treatment associated with endocrine therapy [28]. Several attempts have been made to try to change the formulation of taxanes. Still, to date, nab-paclitaxel is the only formulation able to maintain the same profile of effectiveness but with better tolerability. Nab-paclitaxel has reported comparable efficacy with solvent-based taxanes with the advantage of not needing steroid premedication and a lower infusion time. However, nab-P has been little-studied in older mBC patients [32].

In this context, the NEREIDE study retrospectively assessed the activity and safety data of nab-paclitaxel treatment in 70 women, aged ≥65, with HER2-negative metastatic breast cancer. This is one of the most extensive studies specifically addressing the activity and safety of nab-paclitaxel in elderly patients with MBC; data from previous retrospective analyses were derived from smaller samples [24].

Our analysis, although conducted in a heterogeneous population that, for the most part, had received treatment with nab-paclitaxel as a fourth line, showed promising antitumor activity with a high percentage of objective responses (ORR 31.3%), which favorably compares to that reported in the post hoc analysis of the phase III study with 3-weekly nab-paclitaxel in pretreated patients (27%) [22]. Median PFS and OS reported in the present study are of interest and further underline the importance of this agent in the therapeutic armamentarium of HER2-negative metastatic breast cancer, including elderly and taxane-pretreated patients. Additionally, the safety profile reported here is intriguing, and it is in line with previous reports in younger populations, with no detriment in the quality of life, an extremely important factor in this subset of patients. Furthermore, this data favorably compares to historical data with paclitaxel in elderly patients and, although no formal comparison was performed in our study due to the single-arm design, demonstrate a better safety profile than weekly paclitaxel with approximately one third of the patients discontinuing therapy due to fatigue [33]. These results further support the safe use of nab-P in this population. Our study has some limitations. First, the retrospective nature of the study. In addition, the use of weekly nab-P was reported only in a portion of the patients, limiting direct comparisons between the two schedules of treatment. Moreover, the study was conducted in a pre-immunotherapy era, and no treatment-naïve patients were included in this analysis, and molecular profiling of the tumor was not routinely performed at the time, precluding further subgroup analyses on the mutational status of selected genes, such as *PIK3CA* mutation, *PTEN* loss, etc.

Finally, data reported in our study further support the preferential use of weekly nab-paclitaxel in the elderly, given the more favorable toxicity profile compared with the 3-weekly schedule, as previously reported [24,25].

## 5. Conclusions

The results of the NEREIDE study further confirm, in a real-world setting, the efficacy and safety of single-agent nab-P in pretreated, elderly, HER2-negative mBC. Despite the dramatic advancements made over the last few years in the therapeutic management of HER2-negative mBC, nab-P is still a valid therapeutic option. Nowadays, it is also used in combination with immunotherapy in TNBC patients. Our results are noteworthy as the population included in the present study was heavily pretreated with approximately one third of the patients receiving nab-P as ≥4 line of systemic therapy for metastatic disease, highlighting the relatively safe toxicity profile of this agent. Albeit the present study was conducted in the pre-immunotherapy era, these results still have clinical implications and support the safe use of this agent in elderly mBC patients, an often-underrepresented population in clinical trials. Further studies with nab-P in the elderly population are warranted, including immunotherapy-based combinations.

## Figures and Tables

**Figure 1 cancers-17-02069-f001:**
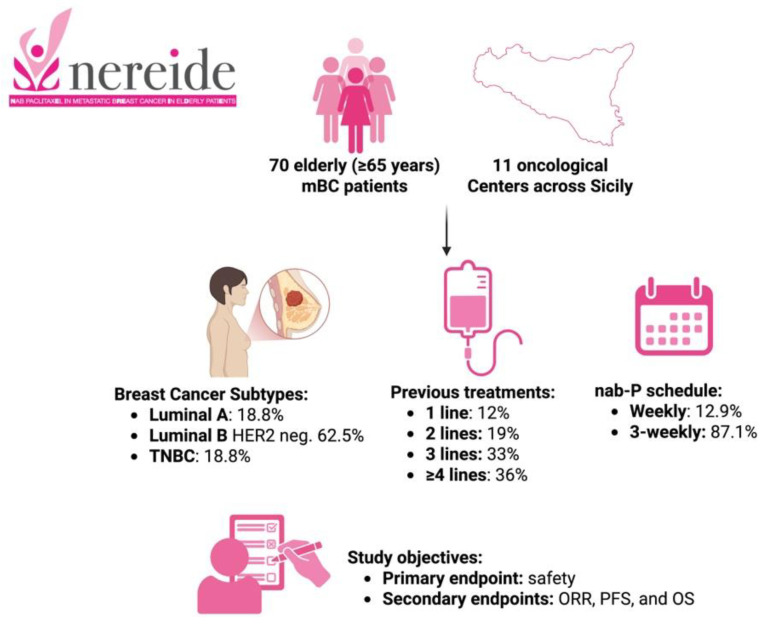
Study design and major patient characteristics (Credit: Created with BioRender.com (https://www.biorender.com)).

**Figure 2 cancers-17-02069-f002:**
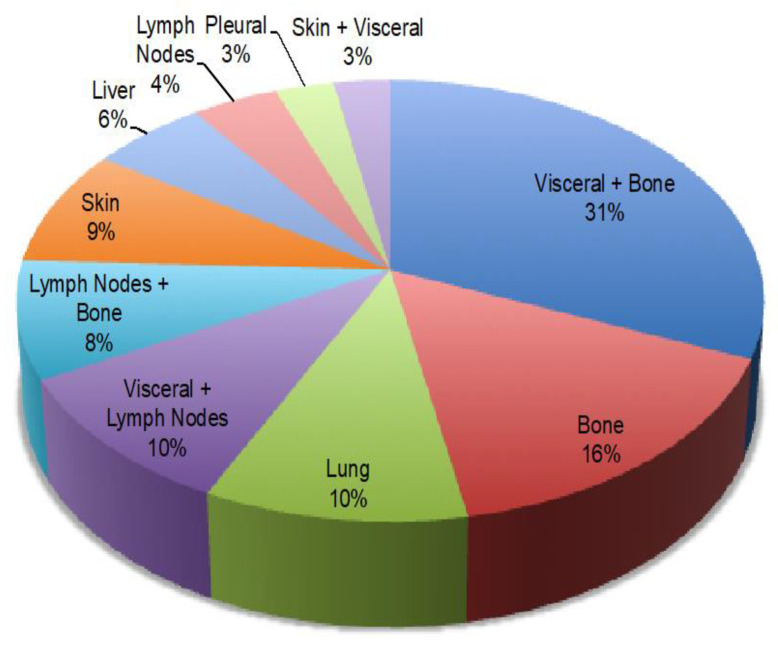
Metastatic sites distribution at baseline in the overall study population.

**Figure 3 cancers-17-02069-f003:**
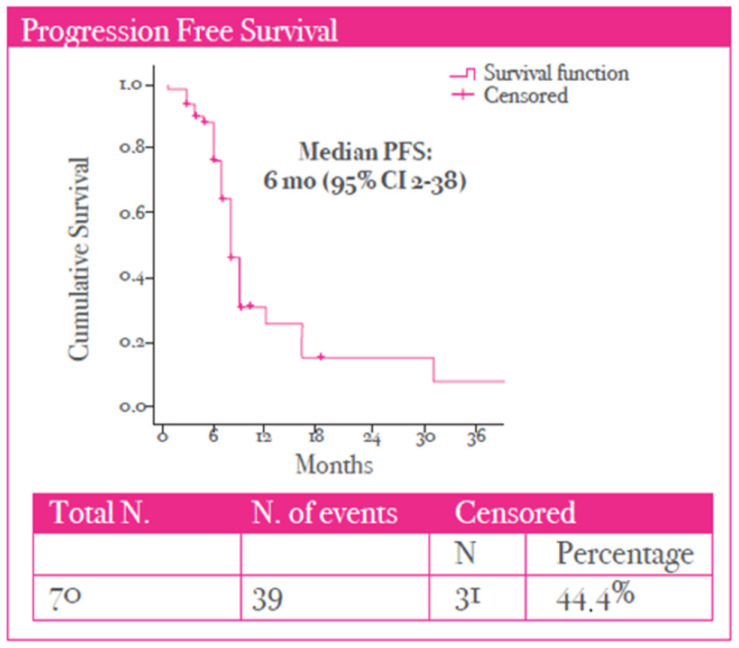
Progression-free survival (PFS) curve in the study population.

**Figure 4 cancers-17-02069-f004:**
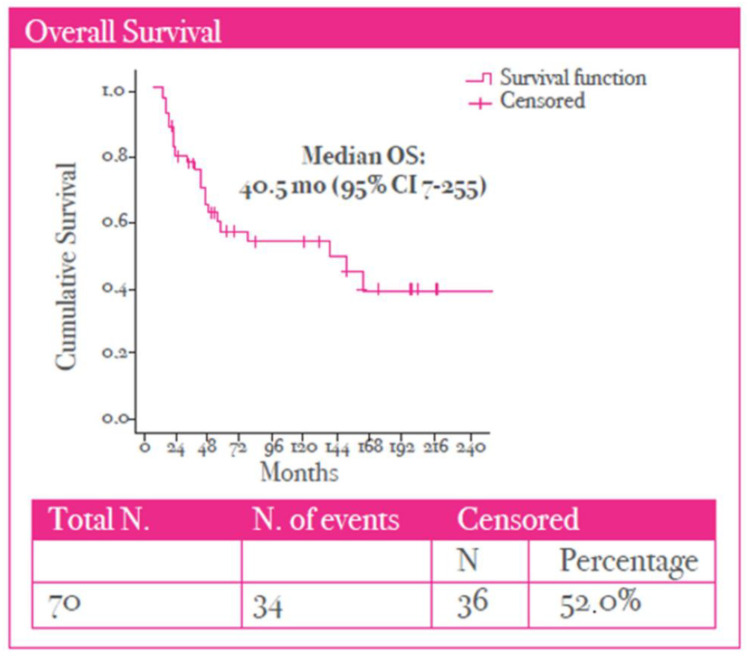
Kaplan–Meier curve for overall survival (OS) for elderly patients treated with nab-paclitaxel.

**Table 1 cancers-17-02069-t001:** Baseline characteristics of the study cohort.

Patients, n	70
Median age (years), range	67 (65–83)
Breast cancer subtypes	Luminal A, 18.75%
	Luminal B HER2-negative, 62.5%
	Triple-negative, 18.75%
Metastatic site distribution	Visceral + bone, 31.4%
	Bone, 15.7%
	Lung, 10%
	Visceral and lymph nodes, 10%
	Lymph nodes and bones, 8.6%
	Liver, 5.7%
	Others, 10%
Number of lines	1 line, 12%
	2 lines, 19%
	3 lines, 33%
	≥4 lines, 36%
Nab-paclitaxel schedule	Weekly, 12.9%
	3-weekly, 87.1%

**Table 2 cancers-17-02069-t002:** Safety profile of nab-paclitaxel in our study cohort.

Overview of Adverse Events	N (%)
Dose reduction	35.5
Treatment discontinuation (%)	11.5
**Major adverse events (%)**	
Fatigue	61.5
Neuropathy	53.8
Leukopenia	39.1
Nausea	21.9
Diarrhea	17.2
Anemia	14.1
Thrombocytopenia	12.5

## Data Availability

Data that support the findings of this study are available on request from the corresponding author.

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
