# Peer review of "Efficacy and Safety Analysis of Nab-Paclitaxel Treatment in Elderly Patients with HER-2 Negative Metastatic Breast Cancer: NEREIDE Study"

_cancers, 2025, doi:10.3390/cancers17132069_

Round 1

Reviewer 1 Report

Comments and Suggestions for Authors

In this retrospective observational study, NEREIDE (Nab-paclitaxel in mEtastatic bREast cancer In ElDerly patiEnts), the authors aimed to evaluate the efficacy and safety of nab-paclitaxel treatment in elderly patients with HER2-negative metastatic breast cancer.

Seventy patients aged ≥65 years with HER2-negative metastatic breast cancer were selected based on specific inclusion and exclusion criteria and were all treated with nab-paclitaxel. In this cohort, 87.1% received nab-paclitaxel at a dose of 260 mg/m² every three weeks, while 12.9% received a weekly regimen of 125 mg/m². Overall, 80.7% of patients had received prior taxane therapy, either in the neoadjuvant (28.6%), adjuvant (25.7%), or metastatic setting (26.2%). All 70 patients were evaluable for both efficacy and safety analyses.

Comments:

1) The progression-free survival (PFS) and overall survival (OS) data presented in Figures 3 and 4 lack interpretability. For the data to be meaningful, they should be compared head-to-head with a control group of patients who received paclitaxel alone. Without this comparison, the data have limited value.

2) Is there any difference in terms of PFS and OS between the patients who received the weekly dose (125 mg/m²) of nab-paclitaxel and those who received the 260 mg/m² dose every three weeks? This comparison should be included in the analysis.

Author Response

Reviewer # 1

In this retrospective observational study, NEREIDE (Nab-paclitaxel in mEtastatic bREast cancer In ElDerly patiEnts), the authors aimed to evaluate the efficacy and safety of nab-paclitaxel treatment in elderly patients with HER2-negative metastatic breast cancer.

Seventy patients aged ≥65 years with HER2-negative metastatic breast cancer were selected based on specific inclusion and exclusion criteria and were all treated with nab-paclitaxel. In this cohort, 87.1% received nab-paclitaxel at a dose of 260 mg/m² every three weeks, while 12.9% received a weekly regimen of 125 mg/m². Overall, 80.7% of patients had received prior taxane therapy, either in the neoadjuvant (28.6%), adjuvant (25.7%), or metastatic setting (26.2%). All 70 patients were evaluable for both efficacy and safety analyses.

Comments:

1) The progression-free survival (PFS) and overall survival (OS) data presented in Figures 3 and 4 lack interpretability. For the data to be meaningful, they should be compared head-to-head with a control group of patients who received paclitaxel alone. Without this comparison, the data have limited value.

2) Is there any difference in terms of PFS and OS between the patients who received the weekly dose (125 mg/m²) of nab-paclitaxel and those who received the 260 mg/m² dose every three weeks? This comparison should be included in the analysis.

R: We thank the reviewer for his valuable comments and appreciate your valuable assessment of our submitted manuscript. We revised the manuscript according to your suggestions and the changes were included in the revised manuscript. We furthermore highlighted in the Discussion some of the limitations raised by the reviewer.

Reviewer 2 Report

Comments and Suggestions for Authors

In this manuscript, the authors retrospectively evaluated the safety and efficacy of nab-paclitaxel (nab-P) in patients with HER2-negative MBC who are ≥65 years old. The results of the study showed a promising anti-tumor activity with a high percentage of ORR (31.3%), which favorably compares to that reported in the post hoc analysis of the phase III study with 3-weekly nab-P in pretreated patients (27%). Median PFS and OS are also promising and further underline the importance of this agent in the treatment of HER2-negative MBC, including elderly and taxane-pretreated patients. Additionally, the safety profile is favorable. This paper offers valuable insights for treatment decisions in elderly patients with HER2-negative MBC, though several critical issues need to be addressed before considering its acceptance.

Major comments:

  1. The authors should highlight detailed information on SAE (≥G3 AE) in a separate table and provide further discussions if any difference between 260 mg/m2 Q3W and 125 mg/m2 QW dosing schedule.
  2. Subpopulation analysis for MBC patients with alterations PI3K-AKT signaling (PIK3CA mutation, PTEN loss, et al) should be provided if feasible.
  3. The authors should provide the data for DOR (duration of response) if feasible.

Author Response

REVIEWER 2

Comments and Suggestions for Authors

In this manuscript, the authors retrospectively evaluated the safety and efficacy of nab-paclitaxel (nab-P) in patients with HER2-negative MBC who are ≥65 years old. The results of the study showed a promising anti-tumor activity with a high percentage of ORR (31.3%), which favorably compares to that reported in the post hoc analysis of the phase III study with 3-weekly nab-P in pretreated patients (27%). Median PFS and OS are also promising and further underline the importance of this agent in the treatment of HER2-negative MBC, including elderly and taxane-pretreated patients. Additionally, the safety profile is favorable. This paper offers valuable insights for treatment decisions in elderly patients with HER2-negative MBC, though several critical issues need to be addressed before considering its acceptance.

R: We thank the reviewer for his valuable comments and appreciate your valuable assessment of our submitted manuscript.

Major comments:

  1. The authors should highlight detailed information on SAE (≥G3 AE) in a separate table and provide further discussions if any difference between 260 mg/m2 Q3W and 125 mg/m2 QW dosing schedule.

R: Thanks for your comment. We updated safety information in the manuscript, detailing the grade 3 or more AEs. Due to the small number of patients experiencing G≥3 AEs we did not include a separate table for these events.

  1. Subpopulation analysis for MBC patients with alterations PI3K-AKT signaling (PIK3CA mutation, PTEN loss, et al) should be provided if feasible.

R: Thanks for pointing this out. Unfortunately, this information was not collected at the time of study conduction as they were not routinely performed in clinical practice. We included this in the study limitations.

  1. The authors should provide the data for DOR (duration of response) if feasible.

R: Thanks for pointing this out. Unfortunately, this information was not collected at the time of study conduction and we couldn’t include this data in the revised manuscript

We look forward your kind answer.

Yours faithfully,

Giuseppina Rosaria Rita Ricciardi, MD, PhD

Round 2

Reviewer 1 Report

Comments and Suggestions for Authors

The main limitation of the study has been acknowledged by the authors and addressed in the discussion section of  revised manuscript. I have no further comments.

Reviewer 2 Report

Comments and Suggestions for Authors

Following the authors' responses, the manuscript has been sufficiently improved to warrant publication in Cancers